# Teaching the Foundations of Machine Learning with Candy

**Daniela Huppenkothen** [1 2]  **Gwendolyn Eadie** [3 4]

## Abstract

Machine learning is ubiquitous in decision making processes across society. The prevalence of ML drives a need for improved education in key concepts at the secondary and tertiary levels that not only trains people to become informed citizens but also trains future researchers to be both principled and ethical practitioners. In this vein, we present a structured classroom activity that simultaneously teaches both supervised classification and critically thinking about ML applications and ethics. We use an active, object-based learning approach to teach supervised classification using a variety of candies, and a problem-based scenario to encourage critical questions about ethics in ML applications.

## 1. Motivation

Machine learning (ML) has become an integral part of decision-making across virtually all sectors and industries. ML has even been applied in areas where the ethics of its use are under scrutiny — for example, from public health (Ravì et al., 2017) and diagnostic medicine (Suzuki, 2017) to decisions about hiring and promotion (Raghavan et al., 2020). Moreover, the extraordinary pace of both development of new algorithms and their implementation in real-world decision making has left curricula lagging across the board.

At universities, ML is often taught only as part of dedicated computer science or data science degrees, while secondary school curricula are traditionally even slower to adopt new content (though an increasing number of proposals and pro-

grammes exist, e.g. Gavalda 2008; Ali et al. 2019). At the same time, an ecosystem of for-profit and non-profit "boot-camps", massive online open courses (MOOCs) and online tutorials have aimed to democratize education in ML in a more ad-hoc, haphazard fashion, leaving both learners and teachers with a wide range of materials and little guidance for its use. In addition, ML materials often stress one of two extremes: theoretical foundations or immediate application.

Teaching materials that focus on very theoretical foundations require a strong mathematical background and may not suit learners with less quantitative education experience. For example, the engineering and computer science curricula at the University of Florida contains two senior-level courses "Current Topics in Machine Learning" in their *Sustainable Model for Assimilating Research and Teaching* undergraduate program (Georgiopoulos et al., 2009). In these courses, students learn from and work with multiple faculty members whose research areas use machine-learning techniques. Prior to these courses, students are also provided with introductory modules on ML in their sophomore and junior years. Although the curricula provides a well-structured program for engineering and computer science students, the teaching materials and approach may be less accessible to people outside these fields. On the other extreme, there are ML materials that aim to quickly bring novice practitioners to a level where they can apply popular algorithms in a short time[1] . But this rushed application pays a price — the practitioners may use ML largely as a black box.

We believe there is a need for and interest in teaching machine learning with the twin objectives: (1) providing learners with the tools and understanding to become the next generation of researchers and practitioners, and (2) training learners to critically evaluate the impact that ML has on their and others' lives. The literature on best practices in teaching machine learning are still new, but as Sulmont et al. (2019) found through an exploratory study, "Student preconceptions include ideas that ML is important, but also not accessible. [...] Instructors reported students having difficulty appreciating the human decision-making aspects of ML, and overestimating the power of ML to solve real-world problems." Both of these findings speak to our twin

---

[1]DIRAC Institute, Department of Astronomy, University of Washington, 3910 15th Ave NE, Seattle, WA 98105, USA [2]The University of Washington eScience Institute, The Washington Research Foundation Data Science Studio, University of Washington, Seattle, WA 98105, USA [3]David A. Dunlap Department of Astronomy & Astrophysics, University of Toronto, Toronto, ON, Canada [4]Department of Statistical Sciences, University of Toronto, Toronto, ON, Canada. Correspondence to: Daniela Huppenkothen <dhuppenk@uw.edu>.

*Proceedings of the 35th International Conference on Machine Learning*, Stockholm, Sweden, PMLR 80, 2018. Copyright 2018 by the author(s).

---

[1]e.g. https://machinelearningmastery.com/machine-learning-in-python-step-by-step/

objectives.

Creative and fun learning activities for teaching machine learning also exist. These approaches may gloss over the mathematical foundations in order to stress conceptual understanding. For example, Rattadilok et al. (2018) present a gamified approach to teaching ML through use of a war-type game called "Clash of Clans". Learners are provided with different machine learning techniques that automate game-play, and are asked to test their ability to mimic the outcome of novice, intermediate, and expert human players. The authors argue that the real-time experience of using these techniques provides another avenue for learners to explore the concepts of ML without mathematical rigor.

Education research suggests that *fun* in the delivery of instruction is positively related to learner engagement (Tews et al., 2015; 2017), and is identified as a motivator to attend classes and seek out new skills (Lucardie, 2014). In addition, *object-based learning*, one of numerous *active learning* approaches, involves the active integration of objects into the learning environment in order to facilitate deep learning and stimulate the learner's imagination (Hannan et al.; Chatterjee & Hannan, 2016). In the field of statistics, there are many active-learning strategies that combine both fun and object-based learning by engaging learners with candy and chocolates (e.g., SherriJoyce & Alexander, 1994; Alexander & SherriJoyce, 1994a;b; Dyck & Gee, 1998; Richardson & Haller, 2002; Lin & Sanders, 2006; Downey, 2013; Schwartz, 2013; Froelich & Stephenson, 2013; Eadie et al., 2019).

Thus, we were motivated to design a creative and fun activity that meets a balance between the two aforementioned extremes of theoretical foundations and immediate applicability of machine learning. Moreover, we sought to instill critical thinking skills about the ethics of using ML techniques in real-life situations. We aimed to develop a modular unit that would serve three distinct target audiences: scientists, including graduate students, with a practical need for machine learning in their research, undergraduate students in different scientific domains where machine learning is taught as part of a larger data analytics units, and students in upper-level secondary education. The development of our unit was motivated by three core questions we keep encountering in our teaching practice:

- ML is often framed in the popular discourse as the domain of mathematicians and computer scientists, which can make it intimidating to individuals without training in these fields. As ML practitioners and teachers, how can we demystify ML to a diverse audience of learners with varying levels of quantitative background?

- Given the prevalence of ML in the world, how can we encourage learners to engage with and understand how ML affects their own lives? Can we put their experiences with ML into the context of the larger, evolving societal discussion around the use of computation in human decision making?

- Machine learning is now easy to use and implement: commercial platforms have graphical user interfaces that abstract away much of the underpinning mathematical concepts. How can we encourage learners without a strong quantitative background to engage with the most important concepts and principles in ML? How can we enable them to make good (and ethical) decisions in the process, rather than use these technologies as a pure black box?

In this paper, we cover the approach, the learning outcomes, and also supply the materials to implement the activity in class.

## 2. Unit Data and Design: Candy, Concepts, and Critical Thinking

Our goal is to provide a hands-on, modular tutorial following The Carpentries' principles for lesson design[2]: working back from key concepts and learning outcomes, we designed a number of challenges that allow learners to engage with (and in some instances, self-discover) concepts important to designing and implementing ML models for real-world decision making.

The emphasis of our ML activity is on building intuition rather than mathematical rigor in a modular, hands-on design. At the same time, we include some mathematical concepts and advanced challenges appropriate for an audience with early college-level mathematics and programming knowledge. We envision this unit – enough content for about three to four hours, depending on depth and the audience's mathematical foundations – to be embedded in a larger formal (data) science curriculum and spread over multiple lessons, or used as a self-contained unit in a Carpentries-style workshop or summer school.

Many ML tutorials and materials are based around well-known data sets like the Iris data set (Anderson, 1936; Fisher, 1936). However, these data sets, while well-understood and curated, were often developed with algorithm bench-marking in mind, rather than teaching. The ability and speed with which learners pick up new concepts is correlated with the familiarity of the components with which these concepts are taught (e.g. Reder et al., 2016). Beginners may find it especially difficult to build their knowledge base about ML with these existing, standard data sets.

---

[2] https://carpentries.github.io/curriculum-development/

Instead of using a well-known training set, we built the unit around a problem wherein the learners must classify different types of candy (see also Figure 1). While we admit the problem is somewhat contrived, it is much closer to our learners' lived realities than many standard data sets. In addition, we believe the involvement of candies helps to establish a welcoming, non-threatening learning environment by introducing a playful character to the lesson design. It also allows learners to physically interact with (and, on occasion, eat) their training samples. While we were initially skeptical that this design would work well for learners beyond the undergraduate level, we have in practice found that individuals of all ages seem to enjoy chocolate-based educational experiences.

Overall, the unit emphasizes the *process* of ML over individual algorithms, which are often the focus of traditional computer science-based curricula. We place particular importance on encouraging learners to ask critical questions about the real-life outcomes of computer-supported decision making early in the process of interacting with or designing a ML system, and keep asking these questions throughout the process. In particular, we place an episode on ethical considerations in building and producing models early in the unit, and refer back to it throughout the remainder of the unit. In this context, our design also dedicates significant discussion time to the process of constructing features for classification, and on evaluation metrics. Overall, we designed the unit to generate an engaging interplay between taught components and self-discovery of important concepts via challenges.

### 2.1. Overall Intended Learning Outcomes

By the end of the unit, learners should be able to ...

- extract meaningful features from data sets, visualize these features, and draw decision boundaries between classes,

- describe how k-nearest neighbours and (optionally) logistic regression classify samples and draw decision boundaries, and apply accuracy as an evaluation metric while keeping its limitations in mind, use cross-validation to test algorithm performance,

- understand that ML is not a magical black box — it is a set of well-motivated, but limited mathematical principles for modelling data sets and predicting future instances using those models, and

- critically interrogate the ML algorithms with which they implement or interact.

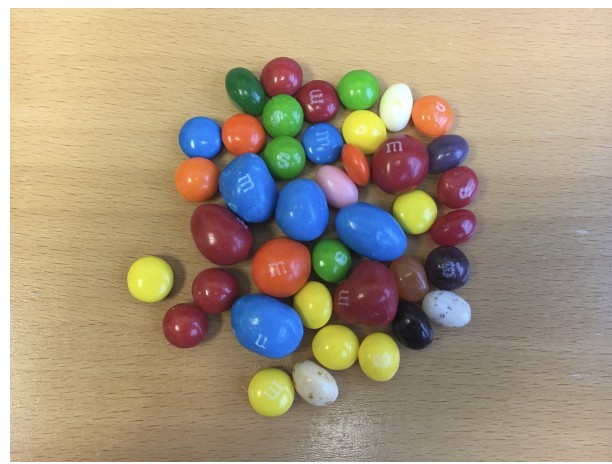

*Figure 1.* An example illustration from the unit depicting different types of candies, used to set up the problem and to engage learners in the process of feature engineering. In in-person contexts or virtual contexts where learners have candies available, they may interact with these objects directly. In other virtual contexts, the images and tables provided with the online materials may be used as stand-ins.

## 3. Unit Outline

The unit starts with a problem statement: *You are preparing your house for a party and have a giant box of mixed candies for snacks, but you have just discovered that one of your guests has a peanut allergy. Therefore, you decide to separate out the candies that contain peanuts and put them in a separate bowl. However, the immensity of sorting through thousands of individual candies by hand is both daunting and impractical. Your friend suggests you use a ML algorithm to sort the candy.*

The above problem statement sets up the discussion around ML as a tool for classification, and leads into the larger discussion around ethical considerations in ML. The next step in the unit guides learners through hands-on exercises with sets of candies, which can be carried out either with real, widely available types of candies of broadly similar shapes and colours (here, we use different types of M&Ms along with Skittles and Jellybeans), if classes are held in an in-person context where these are available, or with images and pre-recorded data sets available with our teaching materials (see e.g. Figure 1). In a context where real candies are available, learners can interact with the different types of candy to self-discover the process of feature-engineering and drawing decision boundaries.

We introduce K-nearest neighbours as one of the most accessible algorithms (see e.g. Figure 2 for an illustration), as well as logistic regression as an optional and somewhat more advanced method. In the context of these algorithms, learners explore accuracy and its limits as a tool for measur-

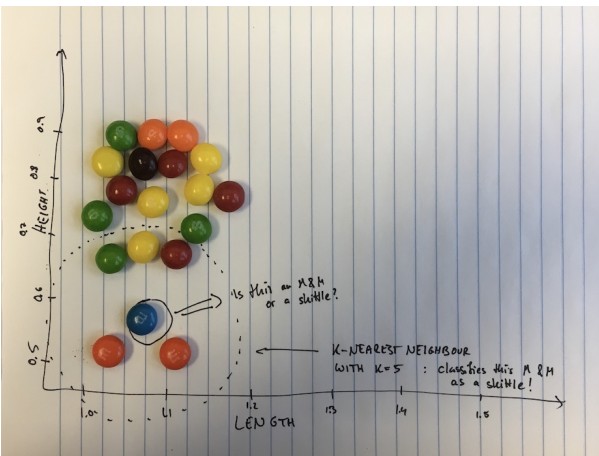

*Figure 2.* Example illustration from a hands-on exercise in the K-Nearest Neighbours episode: learners place candies on a graph of features, approximately according to their feature measurements, and subsequently perform the K-nearest neighbour algorithm by hand on a small set of new candies. This illustration also demonstrates the effect of imbalanced training data on classification results.

ing performance, the effects of imbalanced training data on outcomes, and cross-validation. We follow The Carpentries design principles as much as possible by breaking the unit into modular episodes, each with at most two key concepts and key learning outcomes (though some include optional more advanced materials and challenges).

### 3.1. Episodes and Intended Learning Outcomes

Below we list the episodes for the unit and their intended learning outcomes.

1. **Introduction**: At the end of this episode, learners should have a working understanding of ML, and be able to identify problems where ML may be applied.

2. **Problem Set-up: Classifying Candies**: Learners should understand the classification problem to be solved

3. **Incorporating Ethics into your ML Project**: At the end of this episode, learners understand that ethical considerations are integral to all ML applications, and are able to critically question their own motives and approach to solving their problem. Learners can also lay out worst-case scenarios in order to design effective counter-measures to misuse.

4. **Feature Engineering**: Learners discover that in order to classify objects, many approaches require well-structured information. They can identify features from a set of objects and quantify these features.

5. **Decision Boundaries**: At the end of this episode, learners understand the relevance of different features for the success of classification. They also understand the concept of a decision boundary and how to draw one.

6. **K-Nearest Neighbours**: Learners should be able to describe in qualitative terms the K-Nearest Neighbours algorithm. More advanced learners should be able to write their own version of the algorithm in code, and use the data they generated to classify new data points.

7. **Model Evaluation**: At the end of this episode, learners know how to split training data into training and test sets. Learners can state the differences between true positives, true negatives, false positives, false negatives, can use these definitions to define precision, recall, specifity and F1 score, and understand when they are relevant.

8. **Logistic Regression**: Learners develop an intuition about logistic regression as a way to model two-class problems. More advanced learners can also write their own version of the algorithm in code, and use the data they generated to classify new data points.

9. **Cross-Validation**: Learners can apply cross-validation to their candy data set to test the validity of their algorithm. Learners understand the difference between overfitting and underfitting and the problems they cause.

10. **Final Thoughts**: Learners reflect on the unit, and identify ethical reasons why ML might not be an appropriate tool for this scenario.

### 3.2. Online Materials

Online materials and teaching notes are shared in a public (CC-BY 4.0 licensed) *git* repository, hosted on *GitHub*[3] and rendered via *GitHub Pages*[4], using the Carpentries Lesson Template[5]. The repository and the materials therein are open to contributions, sharing and adaptation for individual instructors' needs. Similarly to the Carpentries' style, the materials aim to both give guidance to the instructor and function as a set of lecture notes that learners can refer back to independently after the unit is completed. Pre-recorded data is shared as part of the repository to enable teaching the unit in contexts where candy are not available (e.g. in virtual settings).

---

[3]https://github.com/dhuppenkothen/machine-learning-tutorial
[4]https://huppenkothen.org/machine-learning-tutorial/
[5]https://carpentries.github.io/lesson-example/

## 4. Conclusion

Given ML's relevance to society, it is important to teach ML to a wide range of learners. Because many educational efforts focus on training *practitioners* and often require a strong quantitative background, we created an introductory unit aimed at teaching important concepts of ML to a variety of learners in a playful, intuitive manner. We envision this work as a resource in the niche between theory-heavy classes and very practical online tutorials. We have implemented several major elements of this tutorial in pilot experiments for an audience of graduate students and post-doctoral fellows at a summer school in theoretical astrophysics, and have taught other methodologies with candy in a range of different post-secondary education contexts. Student feedback suggests that teaching with candy as a simple, real-world example can make complex topics approachable to learners, and reduce learners' apprehension of subject stereotyped as heavily mathematical. In addition, especially when embedded in a longer summer school-type event, learners appreciated the fun and creativity the activities injected into the curriculum as a balance to heavily technical subjects. Learners also valued the interactivity and modularity encouraged of the the Carpentries' teaching model.

## Acknowledgements

We thank the anonymous reviewers for their helpful comments on the draft manuscript. We thank the members of the DIRAC Institute for their help with data collection of candy measurements in the construction of the pre-recorded sample data set. D.H. acknowledges support from the DIRAC Institute in the Department of Astronomy at the University of Washington. The DIRAC Institute is supported through generous gifts from the Charles and Lisa Simonyi Fund for Arts and Sciences, and the Washington Research Foundation. This work was supported by a Data Science Environments project award from the Gordon and Betty Moore Foundation (Award 2013-10-29) and the Alfred P. Sloan Foundation (Award 3835) to the University of Washington eScience Institute and by the eScience Institute.

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
