# OpenReview forum: "Teaching the Foundations of Machine Learning with Candy"
_ECMLPKDD.org/2020/Workshop/TeachML — ECML PKDD 2020 TeachML_

### Official Review · AnonReviewer3 · 2020-07-18
**Well motivated and effective teaching style for all ages.**

**Rating:** 10
**Confidence:** 5

**Review:**

### Summary:
This work addresses the middle ground between math-heavy foundational and application-focused black-box Machine Learning (ML) teaching styles. Authors proposed a fun active learning strategy to understand ML concepts and ethical decision processes using candies. The proposed 10 episode strategy along with their expected learning outcomes are publicly available. Authors found their strategy to be useful for not only for undergraduates but individuals from all ages.

### Strong Points:
* Very well-motivated paper, there is clearly a need for this kind of teaching style to democratize ML
* Proposed strategies along with the learning outcomes are thoughtful and effective
* Prerecorded data is made available with support for the virtual format

### Weak Points:
* Authors mentioned the strategy was effective across all ages. I wonder the effectiveness of teaching with candies for varying age groups. Young children might be too tempted by the candies than actually learning. On the other hand, Older people might get disinterested because they might feel candies are too distracting. Any exploration on the effectiveness of learning across ages will be valuable.

### Other comments:
* Proper use of this teaching style will definitely be effective to instil critical thinking and improve the public perception towards ML as not a black box, I commend the authors on that.
* I would love to see how a few of the episodes are actually described in more detail within the paper but I understand there’s so much information that can be crammed into 4 pages.

---

### Official Review · AnonReviewer1 · 2020-07-27
**Playful ML teaching approach that also focusses on ethics**

**Rating:** 7
**Confidence:** 3

**Review:**

The authors present a self designed introductory teaching lesson of about 3 to 4 hours to convey the principles and workings of some of the fundamental aspects of ML: feature engineering,
KNN and logistic regression as classification methods and performance evaluation in a playful manner using chocolate and candy as a practical toy dataset and additional motivator.

Their approach follows the guidelines of the Carpentries and aims to develop intuition for ML, bridging the gap between the two most dominant ML teaching styles which either are primarily focussed on
theory and might provide a daunting entry barrier on the one hand, or aim at quick implementation and application of ML algorithms without detailed knowledge of the inner workings on the other hand.

I like the hands-on approach suggested here which teaches basic principles by a very plausible example. Also, the authors put strong focus early on to discuss ethical issues of machine learning and encourage lesson participants to think critically about adoption of ML approaches for given problems which is a strong plus in my opinion and often not emphasized enough in existing tutorials.

I would be very curious to learn how lesson participants respond to this approach. Unfortunately, the authors only present their teaching procedure without giving details on how well their approach worked in practice (however that could be measured...), how participants
reacted, what they had problems with, etc. Also I am missing a clear definition of the target audience that course was designed for.
As an additional (but less important) point: as much as I enjoy candy myself and understand the authors picked it for their motivational value, maybe replacing it with a more healthy alternative would be something to think about.

The authors provide their teaching materials as a github repository which is great and allows direct adoption. Unfortunately, author identification became possible through that and the review can no longer be considered completely "blind".
Nevertheless, I suggest accepting the submission since the approach is innovative and will probably invoke interesting discussions among participants.

---

### Official Review · AnonReviewer2 · 2020-07-29
**Fun method to teach machine learning to non-engineering or non-quantitative students**

**Rating:** 6
**Confidence:** 3

**Review:**

Strong points:
1. The motivation section is well-written. The authors explain their motivations, demonstrate a societal need for a solution, and also showcase previous methods used like class of clans.
2. The paper's idea of using candy as a hands-on machine learning teaching method is novel. Overall, the paper communicates a variety of the author's unique perspectives. For example, the authors believe the original Iris dataset was created with benchmarking in mind, not learning. This is another good point.
3. The episodes and intended learning outcomes section is also well organized into specific topics. The topics themselves are also good introductory topics for teaching machine learning.

Weak points:
1. While paper starts strong, the conclusion is weak. Before the conclusion, the author explains the motivations and ideas to teach machine learning using candy. The conclusion simply states there's an ethics component included. While that's important to include, I would have preferred to see the outcomes of using such a teaching method. Without that, this paper feels incomplete.

---

### Decision · Program_Chairs · 2020-07-31

**Decision:**

Accept

**Comment:**

The reviewers agree that this paper will be accepted.

Please register with the conference as soon as possible! See this page for details:
https://ecmlpkdd2020.net/attending/registration/.
Which asks that at least one author per paper registers until July 31, 2020.
We apologize for the very short notice.